# Effectiveness of *Strongyloides* Recombinant IgG Immunoreactive Antigen in Detecting IgG and IgG4 Subclass Antibodies for Diagnosis of Human Strongyloidiasis Using Rapid Immunochromatographic Tests

**DOI:** 10.3390/diagnostics10090615

**Published:** 2020-08-20

**Authors:** Patcharaporn Boonroumkaew, Lakkhana Sadaow, Oranuch Sanpool, Rutchanee Rodpai, Tongjit Thanchomnang, Weeraya Phupiewkham, Pewpan M. Intapan, Wanchai Maleewong

**Affiliations:** 1Department of Parasitology and Excellence in Medical Innovation, and Technology Research Group, Faculty of Medicine, Khon Kaen University, Khon Kaen 40002, Thailand; hamooploy@gmail.com (P.B.); sadaow1986@gmail.com (L.S.); oransa@kku.ac.th (O.S.); rutchanee5020@gmail.com (R.R.); ppoaoy2528@gmail.com (W.P.); pewpan@kku.ac.th (P.M.I.); 2Research and Diagnostic Center for Emerging Infectious Diseases, Mekong Health Science Research Institute, Khon Kaen University, Khon Kaen 40002, Thailand; tthanchomnang@gmail.com; 3Faculty of Medicine, Mahasarakham University, Maha Sarakham 44000, Thailand

**Keywords:** diagnosis, immunochromatographic test, IgG, IgG4, screening test, rapid test, recombinant antigen, strongyloidiasis, *Strongyloides stercoralis*

## Abstract

Human strongyloidiasis is an important soil-transmitted helminthiasis that affects millions worldwide and can develop into fatal systemic strongyloidiasis in immunosuppressed patients. We have developed two new rapid and simple-to-use immunochromatographic test (ICT) kits for rapid serodiagnosis that support stool examination for clinical diagnosis. *Strongyloides stercoralis* recombinant IgG immunoreactive antigen (GenBank: AAB97359.1; rSsIR-based ICT kit) was used for detection of IgG and IgG4 antibodies. The diagnostic efficacy of both kits was evaluated using human serum samples from strongyloidiasis patients, healthy individuals, and those with other parasitosis. At a prevalence of infection of 36.4%, the diagnostic sensitivity, specificity, positive predictive value, negative predictive value, and accuracy of the rSsIR-based IgG ICT kit were 91.7%, 83.8%, 76.4%, 94.6%, and 86.7%, respectively, and those of the rSsIR-based IgG4 ICT kit were 78.3%, 84.8%, 74.6%, 87.3%, and 82.4% respectively. The concordance between the two kits was 89.7%. The recombinant antigen can be produced to an unlimited extent and the kits can be used as point-of-care diagnostic tools and in large-scale surveys in endemic areas throughout tropical regions without necessitating additional facilities or ancillary supplies.

## 1. Introduction

Human strongyloidiasis, which is transmitted through contact with contaminated soil, is an important intestinal parasitic disease that affects approximately 30 to 900 million people globally [1,2,3,4]. *Strongyloides stercoralis* is the main cause of infection, while *Strongyloides fuelleborni* and *Strongyloides fuelleborni kellyi* have also been reported in populations in Africa, Papua New Guinea, and Thailand [1,5]. Asymptomatic carriers can develop hyperinfection if they are immunocompromised, and fatal systemic strongyloidiasis can develop in immunosuppressed patients (i.e., those administered systemic steroids or cytotoxic treatments such as anti-neoplastic agents [6]).

The disease is normally diagnosed through the detection of parasites in stool samples [4,7], while molecular techniques [8,9,10,11,12,13] and serological tests are the potential suitable approach for supportive diagnosis of human strongyloidiasis [14,15,16,17,18,19,20,21,22,23]. However, such methods are time-consuming and require specialized equipment not generally available at the point-of-care (POC) and often only found in advanced laboratories. Although a rapid diagnostic immunochromatographic test (ICT) has recently been developed as a POC tool using somatic *S. stercoralis* larval soluble extract antigen to detect IgG antibodies in human sera [24], the test uses a native antigen. This limits its practicality owing to limitations in the amount of material able to be extracted from parasites and the need to culture the parasites in a laboratory. A recombinant antigen from *S. stercoralis* third-stage larvae called “NIE” [25] has been established as a highly sensitive and specific antigen for antibody detection in the serodiagnosis of human strongyloidiasis [26,27]. In addition, the recombinant antigens, *S. stercoralis* IgG immunoreactive antigen rSsIR [26] and rSs1a [28], also have potential use in the serodiagnosis of human strongyloidiasis. In this study, we used rSsIR (GenBank: AAB97359.1) as an alternative antigen for an immunochromatographic test (ICT) kit and compared the effectiveness of an rSsIR-based IgG ICT kit (for detecting levels of IgG antibody) with an rSsIR-based IgG4 ICT kit (for detecting levels of IgG4 antibody) in the diagnosis of human strongyloidiasis.

## 2. Materials and Methods

### 2.1. Parasite Antigens

The synthesized gene (rSsIR; GenBank no. AF035657.1) at position 1–471 bp was optimized in *Escherichia coli* expression system and constructed into pET43.1a (+) vector from the company (GenScript, Piscataway, NJ, USA). The rSsIR-plasmid was transformed into an *E. coli* JM109 (Novagen, Darmstadt, Germany) cloning host and an *E. coli* Rosetta-gami 2 (DE3) expression host (Novagen). Following this, the sequencing identified a recombinant plasmid, which yielded the in-frame sequence. Expression of N- and C-terminal-fused His-tag rSsIR was induced with 1 mM Isopropyl 1-thio-β-d-galactopyranoside (IPTG) at 33 °C for 24 h. The soluble rSsIR antigen was purified using Ni-NTA His Bind Resin (Novagen) and dialyzed against distilled water containing proteinase inhibitor (cOmplete^TM^ ULTRA Tablets, Mini *EASY*pack Protease Inhibitor Cocktail Tablets, Roche, Basel, Switzerland). The protein concentration of the purified rSsIR protein was determined by Bradford Protein Assay (Bio-Rad Laboratories, Inc., Hercules, CA, USA) and stored at −70 °C before use.

### 2.2. Human Sera

In this study, the serum samples collected from Thailand were divided into three groups: those from strongyloidiasis patients parasitologically confirmed using the agar-plate culture method (*n* = 60) to determine diagnostic sensitivity [29], those from healthy individuals (*n* = 30) who were free from any intestinal protozoa or helminth infection at the time of blood collection confirmed by stool examination [30], and sera infected with other pathogens (*n* = 75; giardiasis (*n* = 5), amoebiasis (*n* = 5), blastocystosis (*n* = 5), hookworm infections (*n* = 5), ascariasis (*n* = 5), trichuriasis (*n* = 5), trichinellosis (*n* = 5), angiostrongyliasis (*n* = 5), gnathostomiasis (*n* = 5), capillariasis (*n* = 5), opisthorchiasis (*n* = 5), fascioliasis (*n* = 5), taeniasis (*n* = 5), cysticercosis (*n* = 5), and sparganosis (*n* = 5)) confirmed using parasitological methods (other than in the cases of cysticercosis, which was diagnosed using computerized tomography and an enzyme-linked immunosorbent assay (ELISA)) [31]. Diagnostic specificity was determined using the 105 serum samples from the latter two groups. Pooled serum samples from strongyloidiasis patients and healthy individuals were used as positive and negative controls, respectively. These sera were provided by the Khon Kaen University Faculty of Medicine frozen sample bank (stored at −70 °C). The precision of each method was determined by performing each test on the same sample on different days; no day-to-day variation was seen when performing during the one-month period. The diagnostic parameters of sensitivity, specificity, and positive and negative predictive values were computed as previously described [32]. The reporting of experiment and data were performed as the criteria of the STARD 2015 list for reporting diagnostic accuracy studies [33]. Ethical clearance for the use of these samples was obtained from the Khon Kaen University Ethics Committee for Human Research (HE611507, approved 2 November 2018) in accordance with the 1964 Helsinki declaration and its later amendments or comparable ethical standards.

### 2.3. Immunochromatographic Test Kit

The rSsIR-based IgG ICT kit was optimized as follows: the test line (T) was coated with 2.0 mg/mL of recombinant SsIR and the control line (C) with 1.0 mg/mL of goat anti-mouse IgG (Lampire Biological Laboratories; 0.1 µL/mm). These were sprayed on a nitrocellulose membrane (Sartorius Stedim Biotech SA, Goettingen, Germany) using an XYZ3210 Dispense Platform (BioDot, Irvine, CA, USA). The colloidal gold-conjugated mouse monoclonal anti-human IgG (Kestrel BioSciences Co., Pathumthani, Thailand) was sprayed onto a glass microfiber filter (GF33; Whatman Schleicher & Schuell, Dassel, Germany) to form the conjugate pad. The serum samples were diluted with sample buffer at a ratio of 1:100, and 5 µL of diluted serum and 100 µL of chromatography buffer were added into the buffer holes marked in Figure 1 as “S” and “B”, respectively. The results were visually interpreted (unaided) at 15 min according to the interpretation card (Figure 1). Red bands appearing at the C line and T line within 15 min indicated a positive result, whereas a red band at only the C line indicated a negative result (Figure 1). The rSsIR-based IgG4 ICT kit was optimized using a method similar to that described above, except that the colloidal gold-conjugated mouse monoclonal anti-Human IgG4 (Invitrogen) was sprayed to form the conjugate pad and the serum samples were diluted with sample buffer at a ratio of 1:5.

## 3. Results

The two kits were evaluated and compared using sera from strongyloidiasis patients (*n* = 60), healthy individuals (*n* = 30), and those infected with other pathogens (Table 1 and Figure 2). Using a cutoff level of ≥1 for the rSsIR-based IgG kit and ≥0.5 for the rSsIR-based IgG4 kit, the two kits yielded positive results for 55 and 47 strongyloidiasis samples, respectively. None of the 30 healthy control sera yielded positive results. Both kits exhibited some cross-reactivity to serum samples from patients with giardiasis, amoebiasis, blastocystosis, hookworm infections, trichinellosis, angiostrongyliasis, capillariasis, fascioliasis, and sparganosis. Gnathostomiasis, taeniasis, and cysticercosis sera also yielded positive results when testing with the rSsIR-based IgG kit (Table 1). The two kits did not differ significantly in this respect (*p* > 0.05; Exact McNemar’s test), with a concordance of 89.7% (148/165; see Table 2). At a prevalence of disease of 36.4% (60/165), the diagnostic sensitivity, specificity, positive predictive value, negative predictive value, and accuracy of the rSsIR-based IgG ICT kit were 91.7%, 83.8%, 76.4%, 94.6%, and 86.7%, respectively, and of the rSsIR-based IgG4 ICT kit were 78.35%, 84.8%, 74.6%, 87.3%, and 82.4%, respectively. Comparative diagnostic values between the two kits using the Receiver Operating Characteristic (ROC) area are shown in Figure 3.

## 4. Discussion

The ICT kit for diagnosis of human strongyloidiasis was recently developed based on IgG antibody detection in human sera against native antigen extracted from *S. stercoralis* larvae [24] The diagnostic sensitivity, specificity, positive predictive value, and negative predictive value are 93.3%, 83.8%, 76.7%, and 95.7%, respectively. This kit has been suggested as a POC test for the screening of asymptomatic *Strongyloides* carriers. Here, we developed two rSsIR-based ICT kits using *S. stercoralis* recombinant IgG immunoreactive antigen (GenBank: AAB97359.1) as the antigen for detecting IgG and IgG4 antibodies, making native antigens unnecessary in the mass production of the serodiagnostic assays. However, the sensitivities of both recombinant-based ICT kits were lower than the kit that use native antigen extracted from *S. stercoralis* larvae [24]. This is possibly owing to the variety and number sera used, incorrectly folded single rSsIR antigenic proteins, lack conformational epitopes, and variation in the duration of infection among the collected samples.

IgG antibody levels are higher in infected patients who are asymptomatic or mildly symptomatic, but lower in patients with severe strongyloidiasis and co-infection with HTLV-1 [34,35]. Because of this, IgG antibody has long been used in the serological diagnosis of human strongyloidiasis [36]. The IgG4 antibody is the IgG subclass that is least present in healthy human serum, accounting for only 3–6% of total IgG [37]. In addition, it has a higher specificity for human strongyloidiasis than total IgG antibody [38]. Previous studies have also found that IgG4 antibody levels were high in younger patients and correlated with re-infection with human schistosomiasis [39,40]. However, in our study, the results from rSsIR-based ICT kits for detection of IgG and IgG4 antibodies did not differ significantly (*p* > 0.05; Table 2). This is possibly owing to variation in the duration of infection and type of human serum samples used for diagnostic evaluation.

Some cross-reactivity with sera from patients with other parasitoses was noted (Table 1). This is not likely to be a real problem in a clinical setting, because the clinical presentations of each of these parasitoses are distinct from those of strongyloidiasis. In addition, some of these patients may have had asymptomatic strongyloidiasis, as sera were collected in a strongyloidiasis-endemic area in northeast Thailand. Further testing is required in areas not endemic for *S. stercoralis* in order to determine the efficacy of these kits. Another factor to consider when evaluating sensitivity is that some strongyloidiasis sera, in which *S. stercoralis* was found using the agar-plate culture method, tested negative (Table 1). This may have been owing to those serum samples having been collected in the acute phase of strongyloidiasis and immunodeficiency of the patient, resulting in the low antibody response.

Recombinant NIE derived from *S. stercoralis* third-stage larvae was used as the antigen for serodiagnosis of human strongyloidiasis, that is, ELISA [26,41] and luciferase immunoprecipitation systems (LIPS) [26]. A previous study found that the sensitivity, specificity, positive predictive value, and negative predictive value of the NIE-ELISA were 97%, 95%, 88%, and 99%, respectively for IgG antibody detection and 45%, 100%, 100%, and 64%, respectively for IgG4 antibody detection [26]. The same study found these values in the NIE-LIPS to be 97%, 100%, 100%, and 99%, respectively for IgG antibody detection and 87%, 100%, 100%, and 90%, respectively, for IgG4 antibody detection [26]. When a combination of recombinant NIE and SsIR antigens was used in a LIPS assay, the sensitivity, specificity, positive predictive value, and negative predictive value were all 100%. Recently, Yunus et al., 2019 [41] used a combination of recombinant NIE and Ss1a [28] antigens to detect IgG4 antibody by lateral flow dipstick test and ELISA and found the sensitivity of both tests to be 91.3 and specificity to be 100%. In the present study, the diagnostic sensitivity, specificity, positive predictive value, and negative predictive value of the rSsIR-based IgG ICT kit for IgG antibody detection were 91.7%, 83.8%, 76.4%, and 94.6%, respectively, and those of the rSsIR-based IgG4 ICT kit for IgG4 antibody detection were 78.3%, 84.8%, 74.6%, and 87.3%, respectively. The ROC area (Figure 3) of the IgG kit was higher than that of the IgG4 kit. The differences in these diagnostic values were owing to differences in the number of evaluated samples. Clinicians in endemic areas should be relied upon for interpretation of results. Moreover, the limitation of this study is the methods were performed based on retrospective data collection, however, the prospective studies on field conditions need to be done in the next observation.

## 5. Conclusions

We were able to successfully develop new diagnostic tools that are fast, simple to use, and can supplement stool examination for clinical diagnosis of strongyloidiasis using recombinant *S. stercoralis* antigen without limits to production. These can be used at the local level in large-scale sero-epidemiological investigations in endemic areas without the necessity for additional facilities or ancillary supplies. This method is important for screening asymptomatic infected individuals and populations who are at risk of developing hyperinfection syndrome or disseminated strongyloidiasis if they are given immunosuppressive treatment for other conditions.

## Figures and Tables

**Figure 1 diagnostics-10-00615-f001:**
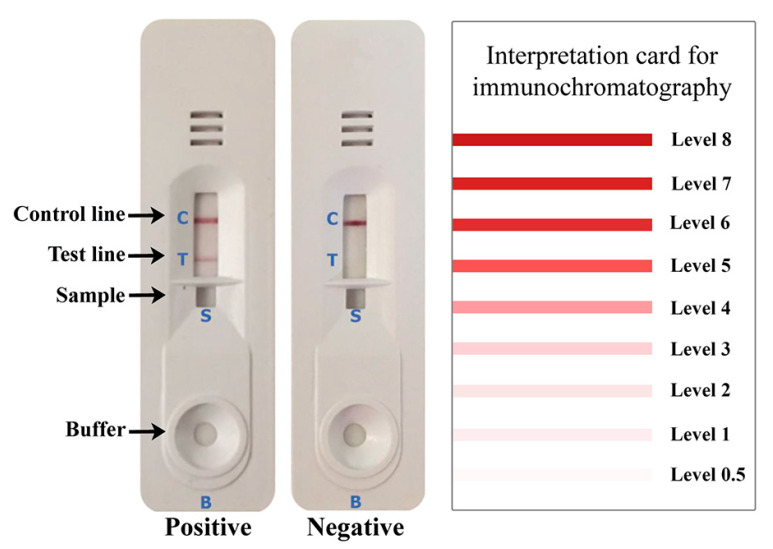
The rSsIR-based immunochromatographic test (ICT) kits for diagnosis of human strongyloidiasis. The intensity of the bands was visually estimated according to the interpretation card. Shown are representative images of positive and negative pooled serum samples. A criterion of the diagnostic result is whether a red band appears at the test (T) line after 15 min. When a serum sample is positive, the T and control (C) lines turn red, whereas only the control line turns red if the serum sample is negative. The cutoff intensity level for a positive result was ≥1 for the rSsIR-based IgG ICT kit and ≥0.5 for the rSsIR-based IgG4 ICT kit. S and B indicate sample and buffer holds.

**Figure 2 diagnostics-10-00615-f002:**
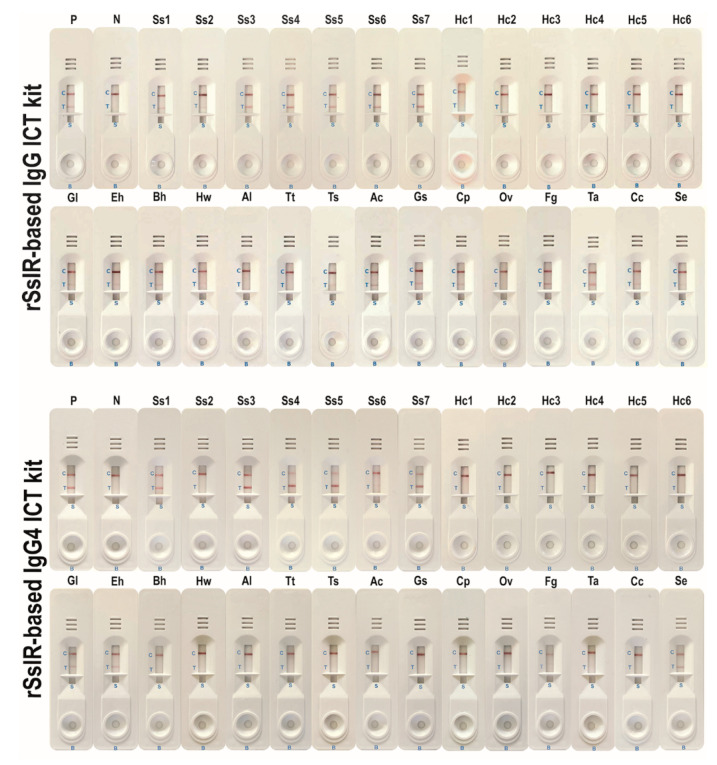
Representative results of the rSsIR-based IgG ICT kit and the rSsIR-based IgG4 ICT kit. P, positive pooled serum samples; N, negative pooled serum samples; Ss1–Ss7, strongyloidiasis; Hc1–Hc6, healthy serum; Gl, giardiasis; Eh, amoebiasis; Bh, blastocystosis; Hw, hookworm infections; Al, ascariasis; Tt, trichuriasis; Ts, trichinellosis; Ac, angiostrongyliasis; Gs, gnathostomiasis; Cp, capillariasis; Ov, opisthorchiasis; Fg, fascioliasis; Ta, taeniasis; Cc, cysticercosis; Se, sparganosis.

**Figure 3 diagnostics-10-00615-f003:**
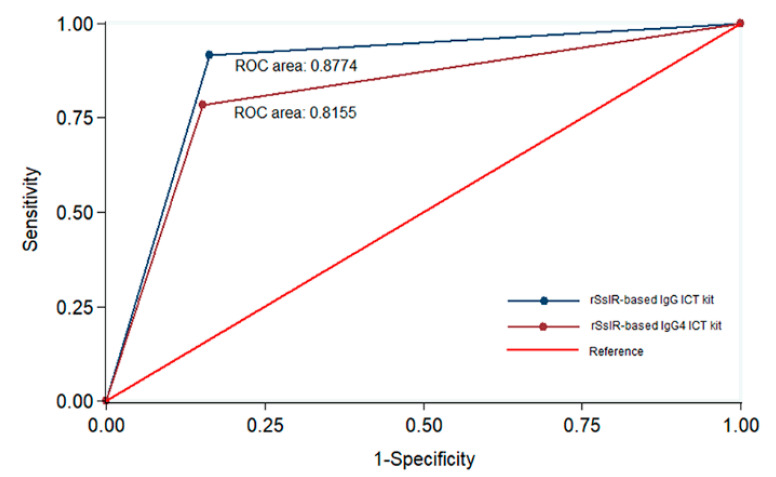
Comparative accuracy between the rSsIR-based IgG ICT kit and the rSsIR-based IgG4 ICT kit. The graph displays sensitivity (true positive rate) of test values on the *Y*-axis and 1-specificity (true negative rate) on the *X*-axis. ROC; Receiver Operating Characteristic.

**Table 1 diagnostics-10-00615-t001:** Results by type of serum sample.

Types		Number of Positive/Total Number of rSsIR-Based IgG ICT Kits	Number of Positive/Total Number of rSsIR-Based IgG4 ICT Kits
Proven strongyloidiasis		55/60	47/60
Healthy controls		0/30	0/30
Other infections			
	Giardiasis	2/5	2/5
	Amoebiasis	2/5	1/5
	Blastocystosis	2/5	1/5
	Hookworm infections	1/5	2/5
	Ascariasis	0/5	0/5
	Trichuriasis	0/5	0/5
	Trichinellosis	2/5	2/5
	Angiostrongyliasis	1/5	2/5
	Gnathostomiasis	1/5	0/5
	Capillariasis	1/5	1/5
	Opisthorchiasis	0/5	0/5
	Fascioliasis	1/5	2/5
	Taeniasis	1/5	0/5
	Cysticercosis	1/5	0/5
	Sparganosis	2/5	3/5
Accuracy (%) [95% CI]		86.7 [80.5–91.5]	82.4 [75.7–87.9]
Sensitivity (%) [95% CI]		91.7 [81.6–97.2]	78.3 [65.8–87.9]
Specificity (%) [95% CI]		83.8 [75.3–90.3]	84.8 [76.4–91.0]
Positive predictive value (%) [95% CI]		76.4 [64.9–85.6]	74.6 [62.1–84.7]
Negative predictive value (%) [95% CI]		94.6 [87.9–98.2]	87.3 [79.2–93.0]

ICT, immunochromatographic test; CI, confidence interval.

**Table 2 diagnostics-10-00615-t002:** Comparison between the rSsIR-based IgG ICT kit and rSsIR-based IgG4 ICT kit.

Test Type and Results	rSsIR-Based IgG4 ICT Kit		
rSsIR-Based IgG ICT Kit	Number of Positive	Number of Negative	Total
**Number of Positive**	59	13	72
**Number of Negative**	4	89	93
**Total**	63	102	165

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
