# Peer review of "Effectiveness of Strongyloides Recombinant IgG Immunoreactive Antigen in Detecting IgG and IgG4 Subclass Antibodies for Diagnosis of Human Strongyloidiasis Using Rapid Immunochromatographic Tests"

_diagnostics, 2020, doi:10.3390/diagnostics10090615_

Round 1
Reviewer 1 Report
Dear editor, the manuscript entitled “Effectiveness of Strongyloides Recombinant IgG Immunoreactive Antigen in Detecting IgG and IgG4 Subclass Antibodies for Diagnosis of Human Strongyloidiasis Using Rapid Immunochromatographic Tests” is an interesting paper evaluating the accuracy of a novel rapid immunochromatographic tests for the diagnosis of human strongyloidosis. This diagnostic study is a case-control study based on laboratory testing of stored human sera. The topic is interesting and the methodology appropriate, the paper is well written. I have some major comments to be addressed by the authors and few minor comments. Firstly, authors should consider, beginning from title, to review the paper and to do appropriate changes, taking into consideration the STARD check list. I also suggest to cite the STARD list recognizing its use for the reporting of experiment and data. Secondly, authors should recognize the limitation of the methodology of this study which is its design itself. In fact studies based on retrospective data collection tends to overestimate diagnostic accuracy (see CMAJ. 2006 Feb 14; 174(4): 469–476. doi: 10.1503/cmaj.050090). Prospective studies on field conditions should be suggested at the end. Finally, the rSsIR-based IgG4 ICT kit shouldn’t to my opinion be presented paired to the other because less interesting in terms of performance. I suggest a comment on the test in a separate way in the discussion, highlighting the more promising tool.
I have few minor comments:
Lines 35-36: epidemiology should refer to the recent paper of Bisanzio et al (doi: 10.3390/pathogens9060468)
Lines 173-174: not only the acute phase could be the problem but also immunodeficiency of the patient, this should be recognized.
Author Response
Point by point response to reviewer 1 comments
Reviewer 1
English language and style
Dear editor, the manuscript entitled “Effectiveness of Strongyloides Recombinant IgG Immunoreactive Antigen in Detecting IgG and IgG4 Subclass Antibodies for Diagnosis of Human Strongyloidiasis Using Rapid Immunochromatographic Tests” is an interesting paper evaluating the accuracy of a novel rapid immunochromatographic tests for the diagnosis of human strongyloidosis. This diagnostic study is a case-control study based on laboratory testing of stored human sera. The topic is interesting and the methodology appropriate, the paper is well written. I have some major comments to be addressed by the authors and few minor comments. Firstly, authors should consider, beginning from title, to review the paper and to do appropriate changes, taking into consideration the STARD check list. I also suggest to cite the STARD list recognizing its use for the reporting of experiment and data. Secondly, authors should recognize the limitation of the methodology of this study which is its design itself. In fact studies based on retrospective data collection tends to overestimate diagnostic accuracy (see CMAJ. 2006 Feb 14; 174(4): 469–476. doi: 10.1503/cmaj.050090). Prospective studies on field conditions should be suggested at the end. Finally, the rSsIR-based IgG4 ICT kit shouldn’t to my opinion be presented paired to the other because less interesting in terms of performance. I suggest a comment on the test in a separate way in the discussion, highlighting the more promising tool.
Reply: we would like to thank for your kind suggestions. Your comments are encouraging and helpful.
The reporting of experiment and data were performed as the criteria of the STARD 2015 list for reporting diagnostic accuracy studies (Cohen JF, Korevaar DA, Altman DG, et al. STARD 2015 guidelines for reporting diagnostic accuracy studies: explanation and elaboration. BMJ Open 2016; 6: e012799. doi:10.1136/bmjopen-2016-012799). For clearer, we added the sentences “The diagnostic parameters of sensitivity, specificity, positive and negative predictive values, were computed as previously described (Galen 1980). The reporting of experiment and data were performed as the criteria of the STARD 2015 list for reporting diagnostic accuracy studies (Cohen et al. 2016).” please see line 93 to 96.
We agree with the reviewer comment that the limitation of the methodology of this study based on retrospective data collection, however, our study is laboratory setting for diagnostic accuracy. The prospective studies on field conditions need to be done in next observation. We added the sentence “. Also, the limitation of this study is the methods were performed based on retrospective data collection, however, the prospective studies on field conditions need to be done in next observation.” Please see lines 201 to 203.
While, the diagnostic values of the rSsIR-based IgG ICT kit are quite higher than the rSsIR-based IgG4 ICT kit, however, there are some human strongyliodiasis serum samples that showed negative results with the rSsIR-based IgG ICT kit but still showed positive results with the rSsIR-based IgG4 ICT kit. Then the aim of including the rSsIR-based IgG4 ICT kit results is for the another option of clinicians in endemic area for consideration.
I have few minor comments:
Lines 35-36: epidemiology should refer to the recent paper of Bisanzio et al (doi: 10.3390/pathogens9060468)
Reply: we made change Ref 2. From Bethony, J.; Brooker, S.; Albonico, M.; Geiger, S.M.; Loukas, A.; Diemert, D.; Hotez, P.J. Soil-transmitted helminth infections: ascariasis, trichuriasis, and hookworm. Lancet. 2006, 367, 1521-1532. https://doi.org/10.1016/S0140-6736(06)68653-4. to Buonfrate, D.; Bisanzio, D.; Giorli, G.; Odermatt, P.; Fürst, T.; Greenaway, C.; French, M.; Reithinger, R.; Gobbi, F.; Montresor, A.; Bisoffi, Z. The Global Prevalence of Strongyloides stercoralis Infection. Pathogens. 2020, 13, 9, 468. doi: 10.3390/pathogens9060468. As you suggested. And modify text from ……approximately 30 to 100 million people globally…. to …. approximately 30 to 900 million people globally., please see revised manuscript line 36.
Lines 173-174: not only the acute phase could be the problem but also immunodeficiency of the patient, this should be recognized.
Reply: we made change to “This may have been due to those serum samples having been collected in the acute phase of strongyloidiasis, and immunodeficiency of the patient, resulting in the low antibody response.” please see revised manuscript lines 181 to 183.
Reviewer 2 Report
This study by Patcharaporn Boonroumkaew and others assessed the performance of the ICT kits produced by themselves that uses recombinant S. stercoralis protein called, rSsIR for serodiagnosis of strongyloidisis.
The results look promising although there may be some cross-reactivity issues with other parasitosis.
The manuscript is well written and I recommend it to be published in Diagnostics after some modifications (mostly additional information).
Lines 24 and 125 Please provide a reference (s) for this prevalence (36.4%). If not reported elsewhere, please give a description how this number was obtained.
Lines 43–44
This sentence wrongfully gives an impression that antigen detection is one of the methods that is normally used for parasite detection, requiring some modification for accuracy.
Line 50
Please remove "novel" because it was 18 years ago when it was novel.
Line 61
AF035657.1 (639bp) contains a polyA signal sequence. Please describe exactly which nucleotide positions in this accession were synthesized. If the entire length (639bp) were synthesized, please describe so.
If the DNA synthesis was done by a company, please include the company information. If it was synthesized by yourself, please indicate so.
Line 65
Please specify which terminus (N- or C-) was His-tagged
Please add more information regarding the induction condition, such as temperature and induction time.
Lines 68–70
"Purification of the solubule ... by the manufacturer" is not necessary because the essential part of this sentence is described in the previous sentence already.
After line 71
Please described how the protein was quantified.
2.2 Human sera
Did all sera come from the same geographical area?
Line 87
How many times the day-to-day variation was assessed?
Line 118
Different cut-offs were used for each two of assays. How these cut-off intensities were determined ?
Author Response
Point by point response to reviewer 2 comments
Comments and Suggestions for Authors
This study by Patcharaporn Boonroumkaew and others assessed the performance of the ICT kits produced by themselves that uses recombinant S. stercoralis protein called, rSsIR for serodiagnosis of strongyloidisis.
The results look promising although there may be some cross-reactivity issues with other parasitosis.
The manuscript is well written, and I recommend it to be published in Diagnostics after some modifications (mostly additional information).
Reply: We would like to thank you for your comments.
Lines 24 and 125 Please provide a reference (s) for this prevalence (36.4%). If not reported elsewhere, please give a description how this number was obtained.
Reply: This is the prevalence of laboratory setting under this condition ((Numbers of strongyloidiasis patient sera (60) / total number of sera (165)) *100) = 36.4%. However, the prevalence, sensitivity, specificity, positive and negative predictive values and accuracy will be changed when numbers of tested serum sample change. We added to “At a prevalence of disease of 36.4% (60/165)” Please see revised manuscript line 133.
Lines 43–44 This sentence wrongfully gives an impression that antigen detection is one of the methods that is normally used for parasite detection, requiring some modification for accuracy.
Reply: We made change to “The disease is normally diagnosed through the detection of parasites in stool samples [4,7], while molecular techniques [8–13] and serological tests are the potential suitable approach for supportive diagnosis of human strongyloidiasis [14–23].” Please see revised manuscript lines 43 to 46.
Line 50 Please remove "novel" because it was 18 years ago when it was novel.
Reply: We deleted as suggested.
Line 61 AF035657.1 (639bp) contains a polyA signal sequence. Please describe exactly which nucleotide positions in this accession were synthesized. If the entire length (639bp) were synthesized, please describe so.
If the DNA synthesis was done by a company, please include the company information. If it was synthesized by yourself, please indicate so.
Reply: We modified to “The synthesized gene (rSsIR; GenBank no. AF035657.1) at position 1-471 bp was optimized in Escherichia coli expression system and constructed into pET43.1a (+) vector from the company (GenScript, Piscataway, NJ, USA).” Pleases see revised manuscript lines 62 to 65.
Line 65 Please specify which terminus (N- or C-) was His-tagged
Please add more information regarding the induction condition, such as temperature and induction time.
Reply: We modified to “Expression of N- and C-terminal-fused His-tagged rSsIR was induced with 1 mM Isopropyl 1-thio-β-D-galactopyranoside (IPTG) at 33 °C for 24 hrs.” Pleases see revised manuscript lines 67 to 69.
Lines 68–70 "Purification of the solubule ... by the manufacturer" is not necessary because the essential part of this sentence is described in the previous sentence already.
Reply: We deleted as suggested.
After line 71 Please described how the protein was quantified.
Reply: We added to “The protein concentration of the purified rSsIR protein was determined by Bradford Protein Assay (Bio-Rad Laboratories, Inc., Hercules, CA, USA) and stored at –70 °C before use.” Pleases see revised manuscript lines 73 to 75.
2.2 Human sera
Did all sera come from the same geographical area?
Reply: All serum samples were collected from Thailand. For clearer, we modified the sentence to “……. the serum samples collected from Thailand were divided……”. Pleases see revised manuscript line 78.
Line 87 How many times the day-to-day variation was assessed?
Reply: The day-to-day variation was assessed during one-month period. We modified the sentence to “The precision of each method was determined by performing each test on the same sample on different days; no day-to-day variation was seen when performing during the one-month period.” Pleases see revised manuscript lines 91 to 93.
Line 118 Different cut-offs were used for each two of assays. How these cut-off intensities were determined?
Reply: We interpreted by the intensity of the bands visually estimated according to the interpretation card (Figure 1). Cut-off criterion of both tests were selected by the optimum diagnostic values (the diagnostic sensitivity, specificity, positive predictive value, negative predictive value, and accuracy).
Finally, we would like to thank the reviewer for your kind comments. The comments are encouragement and helpful. We appreciated very much.